# An Effective Model for Estimating Age in Unaccompanied Minors under the Italian Legal System

**DOI:** 10.3390/healthcare11020224

**Published:** 2023-01-11

**Authors:** Roberto Cameriere, Roberto Scendoni, Luigi Ferrante, Dora Mirtella, Luigi Oncini, Mariano Cingolani

**Affiliations:** 1Department of Medicine and Health Sciences, agEstimation Project, University of Molise, via Cesare Gazzani, 86100 Campobasso, Italy; 2Department of Law, Institute of Legal Medicine, agEstimation Project, University of Macerata, via Don Minzoni 9, 62100 Macerata, Italy; 3Department of Biomedical Sciences and Public Health, Marche Polytechnic University, via Tronto 10/A, 60020 Ancona, Italy; 4Radiology Unit, Hospital of Macerata, via Santa Lucia 2, 62100 Macerata, Italy

**Keywords:** forensic age estimation, hand/wrist X-rays, clavicle X-rays, orthopantomography, open apices, unaccompanied minors

## Abstract

This article presents an effective model for estimating the age of subjects without identification documents, in accordance with Italian legislation covering unaccompanied minors, using instrumental methods recognized by the scientific community for age estimation. A decision-making tree has been developed, in which the first step is a physical examination. If secondary sexual characteristics are fully developed and there are no obvious signs of abnormal growth, dental X-rays are the next step. If the roots of the seven left mandibular teeth between the central incisor and the second molar are completely developed, the focus then moves to the third molar. If the index of the third molar (I_3M_) value is less than 0.08, or if third molars are not assessable, the following step is to study the clavicle which, if fully formed, indicates that the subject is an adult with 99.9% probability; otherwise, the probability is 96%. In all other cases (where the I_3M_ is over 0.08), the probability that the subject has reached 18 years is less than 60%. The research, carried out initially on x-rays of the wrist, teeth and clavicle, highlighted the uselessness of the x-ray of the wrist for determining the age of majority, because in our sample, all subjects with incomplete maturity of hand/wrist bones were under 16 years of age; thus, OPT was necessary anyway. What we propose is a practical, easily feasible, fast, economical, and extremely reliable method, which can be used on Caucasian populations and beyond for multiple forensic purposes.

## 1. Introduction

Age determination of living individuals has become vitally important for the international community, especially in view of the increasing number of persons without acceptable identity documents or with birth certificates suspected of being forged. Such persons may have immigrated illegally or committed crimes; they may have been displaced by war, internal conflict, or natural disasters. In particular, the phenomenon of unaccompanied minors is a very sensitive issue, in reference to both the protection of children’s rights and the difficulties of reaching a resolution in international law [1].

Eighteen years is the most commonly accepted age of full criminal responsibility around the world, according to the UN Convention on the Rights of the Child, which was adopted by world leaders in 1989 [2]. Subsequently, methods of age assessment used on unaccompanied minors who are nationals of third countries were also regulated by the European Union for administrative and social purposes (Council of Europe Resolution of 26 June 1997) [3]. Indeed, accurate age estimation is important for protecting young immigrants whose real age is unknown so that they are not exposed to unfair disadvantages.

Recent global estimates indicate that there were around 281 million international migrants in the world in 2020, which equates to 3.6% of the global population [4]. In this context, the number of unaccompanied minor refugees (UMRs), unaccompanied asylum-seeking children (UASC), and separated children has also increased [5]. A direct consequence of the abovementioned problem is an increased need for harmonized and precise scientific methods for confirming age or obtaining a more reliable age assessment. The modern approach to forensic medical age assessment for UASCs is based on the use of clinical data relating to signs of physical (physiological) development, the condition of teeth, and bone development. Protocol for assessing the age of the living has been adopted by the European community [6], and recommendations for using modern data visualization methods are regularly updated [7]. Nevertheless, in the international context, there is certainly no uniform approach to age estimation.

Previous studies have mainly aimed at evaluating the relationship between chronological age and the ratio between the measurements of some anatomical regions, such as the bones of the hand/wrist [8,9,10], the medial clavicular epiphyseal cartilage [11,12,13], and teeth [14,15]. Compared with other parts of the body, images of these structures can be obtained with low levels of radiation; the radiographic positioning is simple; and a significant number of bones or teeth are generally available.

Regarding the medial clavicular epiphysis, computed tomography (CT) is the best choice of imaging modality, according to studies by several authors [16]; however, it is both ethically (exposure to excessive doses of ionizing radiation) and financially difficult to propose studies based on the use of CT. Recently, studies on clavicular ossification have been carried out using magnetic resonance imaging (MRI) [17,18]. In forensic age estimation, there is growing interest in using MRI to avoid exposure to ionizing radiation, and over the last decade much research has been conducted on MRI techniques for visualizing and analyzing different anatomical categories [19,20]. Although the procedure has a high resolution, it incurs higher costs than X-rays, and other limitations are related to the longer execution times and the presence of possible artifacts.

Over the past two decades, within the framework of the AgEstimation Project undertaken by the department of anthropology at the Institute of Legal Medicine, University of Macerata (Italy), several papers have been published on the challenges of age estimation for forensic purposes in young subjects, including methods involving both teeth and hand/wrist bones [21,22,23].

The aim of this research article is to provide guidance on how to select the most useful radiographs for determining 18 years of age. We have created an easy-to-use flow chart to ascertain whether or not a subject is 18 years old, intended for unaccompanied minors and in keeping with Italian legislation. Here we introduce diagnostic imaging tools to help resolve age-related doubts and disputes with ease and scientific reliability.

## 2. Materials and Methods

Several samples were studied to organize the flow chart: 234 hand/wrist X-rays (125 males, 109 females), 962 orthopantomographs (OPTs) (449 males, 513 females) and 203 X-rays (105 males, 98 females) of the medial clavicular epiphyseal cartilage. All samples were obtained from Caucasian subjects aged between 14 and 24 years (Table 1). OPTs were collected from private practice dentists, while X-rays of hands/wrists and clavicles came from public hospital. All the exported images were anonymized.

The chronological age of each subject was calculated by subtracting the date of birth from the date on which the radiograph was carried out for that individual.

Radiographs were either in digital form or digitized using a scanner, and images were recorded on computer files, processed by a computer-aided drafting program (Adobe Photoshop 7). In regard to the hand, complete maturation was evaluated according to Cameriere’s method [24]. For hand/wrist bones, X-rays were taken of the left hand in the postero-anterior projection, with fingers extended and slightly spread apart. The carpal area and the ulnar and radial epiphyses were identified and measured (Ca). The areas of the eight carpal bones were calculated and added together to yield the global values of bone areas (Bo). If two bones overlapped, the common area was calculated only once. Finally, to normalize measurements, the Bo/Ca ratio was calculated.

Tooth maturation was also evaluated, distinguishing the third molar from the other teeth. This was because, by the age of 16, most of the first seven teeth have reached dental maturity, and we therefore simply ascertained whether or not they were completely developed. The only teeth that continue to grow after the age of 18 are the third molars, and we used Cameriere’s third molar maturity index (I_3M_) to assess the age of majority [25].

According to the third molar maturity index, if the root development of the third molar is complete, i.e., if the apical ends of the roots are completely closed, then I_3M_ = 0; otherwise, I_3M_ equals the sum of the distances between the inner sides of the two open apices divided by the tooth length. Both impacted and non-impacted third molars were included in the study, provided that their roots were radiographically distinguishable.

Lastly, the degree of ossification of the medial clavicular epiphyseal cartilage was defined following Schmeling’s classification, according to which there are five phases of ossification [11]. Stages 1, 2, 4 and 5 can be used for adult age estimation, as follows:Stage 1: non-ossified epiphysis;Stage 2: discernible ossification center;Stage 3: partial fusion;Stage 4: the epiphyseal cartilage is fully ossified;Stage 5: the epiphyseal cartilage has fused completely, and the epiphyseal scar is no longer visible.

Physical examination is indispensable to rule out any visible signs of age-related diseases and to establish whether a subject shows incomplete secondary sexual characteristics. Changes in secondary sexual characteristics occur over the course of a few years (2–5 in general), from the onset of puberty to maturity through five stages, as described by Tanner (1962) and commonly accepted, although there are also some approximate definitions. Tanner proposed five stages for each of two sexual characteristics (growth of pubic hair and genital development in males, growth of pubic hair and breast development in females) [26]. Stage 5 corresponds to full sexual maturity. Many studies have been conducted on various populations with regard to the appearance of secondary sexual characteristics according to Tanner’s stages [27].

## 3. Results

This section describes the results of the sample examined, which will allow for the best application of the method, as explained in the Section 4. The first step in the process of assessing a young adult’s degree of biological maturity is to carry out a physical examination, taking into account body weight and height, constitutional type, and sexual maturity. The last measure may be evaluated in various ways in males and females, even though it is highly variable. If secondary sexual characteristics are completely developed, the next procedure in forensic age assessment is usually to examine X-rays of the hand/wrist. In our sample, all subjects who did not have completely mature hand/wrist bones were indeed under 16 years of age. However, if X-rays showed that hand/wrist development was complete, we did not know if he/she was a minor of 16 or 17 years of age, or if he/she has reached the age of majority; the analysis of the seven left mandibular teeth was always necessary at this point. Again, if all seven teeth were not completely mature, the subject was considered to be a minor. In our sample, any subject who was found to have at least one of the seven left mandibular teeth not completely mature was indeed under 16.

If the development of all seven teeth was complete, we then examined the third molar index, I_3M_. However, in our sample, 6% of subjects did not have third molars, due to agenesis, extraction, rotation, or irregularities in tooth position; consequently, they were not assessable radiographically.

According to Cameriere’s I_3M_, if the value of I_3M_ is less than 0.08 or the third molar cannot be evaluated, ossification of the medial clavicular epiphyseal cartilage is assessed by X-ray. If the clavicle is in stages 4 or 5 of Schmeling’s classification, the subject is considered to have reached the age of majority (18 years) with a probability of more than 99.9%; if the clavicle is at a lower stage, the probability is 96%. In addition, if the I_3M_ is over 0.08 but less than 0.15, the probability that the subject is over 18 is 60%. Lastly, if the I_3M_ is equal to or more than 0.15, this probability falls to 16%.

## 4. Discussion

Being able to differentiate between adults and minors by using age estimation techniques is an important issue in the forensic field. According to Directive 2013/33/EU of the EU Parliament and Council [28], which lays down standards for the reception of applicants for international protection, an “unaccompanied minor” is a third-country national or a stateless person under eighteen years of age, who arrives on the territory of the Member State unaccompanied by an adult responsible for him/her whether by law or by the practice of the Member State concerned, and for as long as he or she is not effectively taken into the care of such a person; it includes a minor who is left unaccompanied after he/she has entered the territory of the Member State.

In Italy, the D.P.C.M. no. 234 of 2016 [29], which currently represents the most complete source of Italian legislation on age assessment, establishes that:-only where there are well-founded doubts about age and this cannot be ascertained through identification documents (e.g., a passport or another identification document bearing a photograph), the authorities may request the competent judge for the protection of minors to authorize a multidisciplinary age assessment procedure;-this assessment is conducted, in compliance with the best interests of the minor, by a multidisciplinary team at a public health facility, identified by the judge, and includes a social interview, an auxological pediatric examination, and a psychological or neuropsychiatric evaluation, in the presence of a cultural mediator, taking into account the specificities relating to the ethnic and cultural origin of the interested party;-the minor must be adequately informed, with the help of a cultural mediator, about the type of examinations they will undergo, their purposes, and their right to oppose them;-the final report must indicate the estimated age, specifying the margin of error (due to biological variation and the methods used), and the consequent minimum and maximum values of the attributable age must be given;-in cases where, considering the margin of error, doubts remain as to whether the individual is a minor or an adult, the minor age is presumed;-pending the determination of age, the interested party must in any case be considered as a minor for the purpose of immediate access to assistance and protection.

If an X-ray examination of the wrist-hand region has already been carried out, the person concerned, their guardian, or the defense counsel may ask for a copy of the X-ray image in order to avoid subjecting the person to further tests and therefore being exposed to more radiation.

Overestimation of age is very rare, but must be carefully avoided. It is generally linked to endocrine disorders, including precocious puberty, adrenogenital syndrome, and hyperthyroidism. Underestimation of age, which is more common, does not disadvantage the person concerned.

In 2017, new legislation on protection measures for unaccompanied minors (law n. 47) [30] was passed in Italy, with provisions on age assessment procedures; it provides that an age assessment procedure may be ordered only when there are well-founded doubts about the age declared by an alleged minor and it is not possible to verify this with documentary evidence. Lastly, on 9 July 2020, the Italian government drafted a “Multidisciplinary protocol for determining the age of unaccompanied foreign minors”, according to which a social worker (employed by the Italian National Health System or another local authority) must be integrated into the team of experts called upon. This protocol provides that various professionals are involved in a step-by-step approach.

According to the general rule of law in most countries, the principle “*in dubio pro reo*” is regarded as a rule for making legal decisions. In the context of age determination, this means that a subject should be considered an adult only if there is certainty or high probability; at the same time, only the minimum number of tests should be conducted, with particular attention to X-rays. However, although it has been shown that narrative history can provide valuable information about a child’s age when documentation is absent or clearly incorrect [31], mistakes can be made in holistic age assessment. For legal practice, accurate age determination is of paramount importance, as many courts do not accept age intervals. In such circumstances, analytical and instrumental methods capable of providing results that avoid subjective interpretation [32] should be integrated into the age assessment process. Several studies have applied physical examinations, radiographs of hand/wrist bones and teeth, with particular reference to the third molar, and X-rays of the clavicle [33] as necessary steps for determining adult age. The flow chart reported in Figure 1 is presented as a decision-making tree, providing a sequence of steps to be taken.

If the results of this sequence of verifications are uncertain, the probability that a subject has attained the age of majority (18 years) is reported. The flow chart illustrates a rapid, easy-to-use technique for age estimation with the minimum probability of error.

All subjects tested with wrist and teeth not completely developed were minors under 16 years of age. However, if one was 17, it was still necessary to proceed with dental analysis, with the aim to have a major diagnostic definition of minor age. Therefore, the OPT, with which we were able to evaluate both the growth of the first seven teeth and the I_3M_, was necessary anyway. It is consequently unnecessary to make, from the beginning, X-rays of the hand/wrist to determine age, because through the OPT we can extrapolate all the data necessary to reach the appraisal of majority or minor age. If the I_3M_ is not available, the clavicle will be evaluated.

After physical examination, which focuses on excluding clear-cut pathologies responsible for serious changes in growth, the first materials necessary are OPTs. If the roots of the seven left mandibular teeth, between the central incisor and the second molar are not completely developed, i.e., the roots are not fully closed, the subject is under 18 years old [34,35]. Although tooth development depends on biological sex, it should be noted that, in both boys and girls, maturation is complete before the age of 17 years.

Conversely, if the roots of the seven left mandibular teeth are completely closed, an examination of the third molar, if assessable, is the next step. This is because the third molar is the last tooth to mature, although it is sometimes missing due to agenesis or extraction, or it may be impossible to evaluate radiographically due to a particular rotation or abnormal position. If the value of I_3M_ is higher than 0.08 and lower than 0.15, the probability that the person is an adult (over 18) is 60%, falling to 16% if the index is higher than 0.15. If the I_3M_ is below 0.08, the probability that the subject is of adult age is 96%. The accuracy of bone/dental maturity indices for estimating chronological age has already been evaluated and also reported in recent scientific literature [36]. However, for improved probability levels, especially if the third molar is not assessable, X-rays of the clavicle can be studied [37]. If clavicular ossification is at least in stage 4, the probability that the subject is an adult is 99.9%. This is supported by the results of Garamendi et al., who reported that several studies conducted between 1871 and 2007 with samples of known origin showed that the minimum age of fusion of the medial epiphyses of the clavicle is 19 years [13]. Prior to stage 4, the clavicle cannot be used for estimating age of majority, because in the present study, 46% of subjects in this situation were of adult age. In addition, if the I_3M_ is greater than 0.08, studying the clavicle is not useful. In our data, only one subject (2% of the sample) had a mature clavicle and an I_3M_ exceeding 0.08.

As the groups studied in this paper came from Italy, the question arises as to whether the results can be applied to subjects from other geographical areas where no specific studies have been conducted. Many reports indicate that the results of studies relating to skeletal growth may be applicable to different ethnic groups. Unfavorable socio-economic conditions and health deficits may lead to growth delays, and consequently the sensitivity of the tests used in the flow chart decreases and the probability that an adult is declared to be a minor increases. Thus, even though this underestimation in age would not disadvantage the person concerned, health deficits must be considered when the flow chart is used to estimate the age of majority among individuals with a socio-economic status lower than that of the reference population.

## 5. Conclusions

Easier, quicker, cheaper, and more reliable methods of age estimation are essential to all branches of medicine. In our specific case, age estimation in unaccompanied minors for forensic purposes requires a rapid and easily interpretable method [38]. The flow chart presented here provides such a solution, but it requires exposure to radiation, and it is therefore important that children and young people (or those responsible for them) should be properly informed about this. From a scientific viewpoint, the radiation doses applied are low enough to be ethically acceptable. Current non-radiographic methods for ascertaining whether or not a subject is 18 years old are imprecise, since clinical and sexual evaluations cannot be used to make accurate age determinations, especially after the age of 13/14. Applying the model would require effective collaboration with expert, trained radiologists, which is easily feasible, thus ensuring standardized, correct, adequate, documented, and repeatable age estimates, in line with scientific knowledge and reflecting human rights policies with respect to unaccompanied minors. The flow chart can also be applied to other forensic cases in which it must be established with reasonable certainty whether a subject is over or under 18 years old. In various European countries, lawyers, prosecutors and judges frequently ask whether ethnicity may have an effect on age determination, and a more thorough answer can only be given if methods are tested on specific populations of interest [39].

The application of the method obviously involves the use of ionizing radiation, even if in minimal quantities; this could clash with the prohibitions imposed by the authorities of several states on age estimation for non-clinical purposes using x-ray methods. Moreover, in case of evaluation of the minor when we do not have the dental eruption or the third molar available, the model would not be applicable, and it will be necessary to proceed with alternative methods. Finally, it will be necessary to test the validity of the method on a group of non-Caucasian subjects before considering its routine use for forensic purposes.

## Figures and Tables

**Figure 1 healthcare-11-00224-f001:**
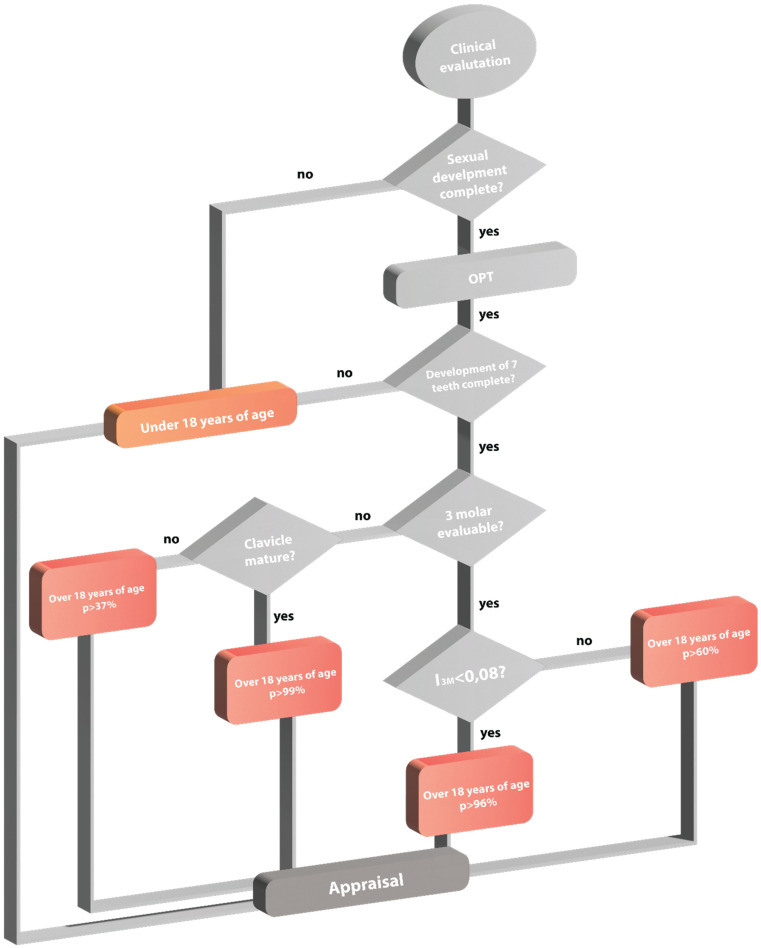
Flow chart of the effective model for estimating the age of unaccompanied minors under the Italian legal framework.

**Table 1 healthcare-11-00224-t001:** Sample distribution according to age, sex, and anatomical region.

Years	Hand		Teeth		3rd Molar		Clavicle	
	M	F	M	F	M	F	M	F
14	12	13	47	49	43	47	10	10
15	11	11	64	66	61	63	12	9
16	16	9	84	79	82	76	10	11
17	15	13	48	69	41	67	13	10
18	17	10	56	68	54	65	11	12
19	15	18	47	64	43	60	10	10
20	10	10	49	49	46	47	13	14
21	10	15	20	31	17	28	14	10
22	19	10	34	38	33	33	12	12
Total	125	109	449	513	420	486	105	98

## Data Availability

The data were obtained from the archives where the subjects underwent clinical-instrumental examinations and are available to the authors.

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
