# Peer review of "An Effective Model for Estimating Age in Unaccompanied Minors under the Italian Legal System"

_healthcare, 2023, doi:10.3390/healthcare11020224_

Round 1

Reviewer 1 Report

The paper is worth of publication. The authors propose a model to define age in young adults. Did they performe a blind check on subjects out of their cohort to demonstrate the overall accuracy of the model? Please specify.

  My comments for improving the paper regard the application of their model to subjects as a check of the correctness of the model. It is not clear if they applied the model to subjects different from the subjects used to prepare the model.

Author Response

-The accuracy of bone/dental maturity indices for estimating chronological age has already been evaluated and also reported in recent scientific literature. We added a recent article in the manuscript, concerning a meta-analysis on the accuracy of the methods that involve bone/dental maturity (Marconi, V.; Iommi, M.; Monachesi, C.; Faragalli, A.; Skrami, E.; Gesuita, R.; Ferrante, L.; Carle, F. Validity of age estimation methods and reproducibility of bone/dental maturity indices for chronological age estimation: a systematic review and meta-analysis of validation studies. Sci. Rep. 2022, 12, 15607). Regarding clavicle stages, the accuracy of the method has been evaluated on CT images (El Morsi D.A.; El-Atta H.M.A.; ElMaadawy M.; Tawfik A.M.; Batouty N.M. Age Estimation from Ossification of the Medial Clavicular Epiphysis by Computed Tomography. Int. J. Morphol. 2015, 33(4), 1419-1426). Indeed, there are no articles that have tested the accuracy of the age estimation method on wrist radiographs, such as the one applied to our study (Bo/Ca). This could be the subject of a future research.

-The subjects of our sample were analyzed and taken into consideration exclusively for the realization of the model. The method was tested, in practice, also in a limited number of unaccompanied foreign minors, for whom it was necessary to verify the age of majority; however, these subjects were not included in our study.

Reviewer 2 Report

Thank you for the invitation to review this paper. Below you will find my comments:

This manuscript provides a concise and comprehensive report on the age determination procedure in legal context in Italy. The step-by-step approach is inspiring and practical. After thorough review, I do have some concerns. 

First, the flow chart for estimating the age of unaccompanied minors under Italian legal framework is straightforward, as in displaying methods orders as well as probability over eighteen with clavicle and third molar maturity. However, examination of hand/wrist X ray is missing from the chart before step OPT, which does not match the text. 

Second, there seem to be a mistake in line 150, If the development of all seven teeth was incomplete, we then examined the third molar index, I3M. Inferred from text, complete of seven teeth formation leads to examination of third molar. 

Third, it’s confusing in line 160-163, In our sample, neither stage 4 nor stage 5 was observed in any subject under the age of 20s. However, these two stages were recorded in only 37% of subjects over the age of 18, and only 54% of stage 4 classifications were attributed to minors.

In my understanding, inference of first and second sentence is contradicting: first sentence indicated that those aged 14-19 were staged 1-3 in clavicle maturity; second sentence indicated that 37% of those aged 18-22 were between stage 4-5. 

Lastly, in the flow chart, probability over eighteen was presented by single method. What about probability over or under eighteen by combination of methods? 

Author Response

This manuscript provides a concise and comprehensive report on the age determination procedure in legal context in Italy. The step-by-step approach is inspiring and practical. After thorough review, I do have some concerns.

1) First, the flow chart for estimating the age of unaccompanied minors under Italian legal framework is straightforward, as in displaying methods orders as well as probability over eighteen with clavicle and third molar maturity. However, examination of hand/wrist X ray is missing from the chart before step OPT, which does not match the text.

In forensic practice the hand/wrist x -ray is still used as the main, and sometimes only, indicator of the age of majority/minority. The study of a sample in this article indicates unequivocally how the use of the wrist is not necessary. In fact we have specified that if one was 17 it was still necessary to proceed with dental analysis, with the aim to have a major diagnostic definition of minor age. So, the OPT, with which we were able to evaluate both the growth of the first seven teeth and the I3M, was necessary anyway. It is consequently useless to make X-rays of the hand/wrist and for this reason it is not used for the final flow chart. We have clarified this aspect better.

2) Second, there seem to be a mistake in line 150, If the development of all seven teeth was incomplete, we then examined the third molar index, I3M. Inferred from text, complete of seven teeth formation leads to examination of third molar.

Thank you for the remark, we have now corrected the mistake. The correct term is “complete”.

3) Third, it’s confusing in line 160-163, In our sample, neither stage 4 nor stage 5 was observed in any subject under the age of 20s. However, these two stages were recorded in only 37% of subjects over the age of 18, and only 54% of stage 4 classifications were attributed to minors.

-In my understanding, inference of first and second sentence is contradicting: first sentence indicated that those aged 14-19 were staged 1-3 in clavicle maturity; second sentence indicated that 37% of those aged 18-22 were between stage 4-5.

Thanks again, there is an error. To avoid misunderstandings we have now decided to remove the entire sentence.

-Lastly, in the flow chart, probability over eighteen was presented by single method. What about probability over or under eighteen by combination of methods?

The clavicle is used for estimation of majority only when the I3M is not evaluable (e.g. agenesis, traumatic loss); therefore a combined accuracy between the two methods (teeth and clavicle) is not needed. The flow chart, in fact, represents a step-by-step approach.

Reviewer 3 Report

I read the article carefully. I believe it is of particular forensic interest. The proposed method is certainly useful and important both for the Italian and international legal situation. The article deserves publication. However, I suggest including limitations of the study that I don't see identified.

Author Response

I read the article carefully. I believe it is of particular forensic interest. The proposed method is certainly useful and important both for the Italian and international legal situation. The article deserves publication. However, I suggest including limitations of the study that I don't see identified.

Thank you for your suggestion. We have now included the following limitations at the end of the Conclusions section.

“The application of the method obviously involves the use of ionizing radiation, even if in minimal quantities; this could clash with the prohibitions imposed by the authorities of several states on age estimation for non-clinical purposes using x-ray methods. Moreover, in case of evaluation of the minor when we do not have the dental eruption or the third molar available, the model would not be applicable and it will be necessary to proceed with alternative methods. Finally, it will be necessary to test the validity of the method on a group of non-Caucasian subjects before considering its routine use for forensic purposes”.